# Probabilistic Feature Smoothed Gaussian Process For Imbalanced Regression

## Abstract

Gaussian Processes (GPs) are non-parametric Bayesian models widely used for regression, classification, and other tasks due to their explainability and versatility. However, GPs face challenges in imbalanced regression, where the skewed distribution of target labels can greatly harm models' performances. In this work, we introduce the Probabilistic Feature Smoothed Partially Independent Training Conditional Approximation (PFS-PITC) to enhance GP performance in imbalanced scenarios. We extract statistical features from the observation space using equidistant label intervals and apply kernel smoothing to address sampling density discontinuities. This process enables PFS-PITC to utilize information from nearby labels within imbalanced datasets, thereby reducing GPs' sensitivity to such imbalances. Empirical tests on various imbalanced regression datasets demonstrate the effectiveness of PFS-PITC, contributing to the robustness of GPs in handling flawed real-world data and expanding their applicability in challenging data processing tasks.

## 1 Introduction

Gaussian processes (GPs) are extensively applied in various machine learning domains, including image classification (Bazi & Melgani (2009); Dutordoir et al. (2020); Xu et al. (2013; 2022)), graph learning (Chen et al. (2022); Miao et al. (2022)), black-box optimization (Koza et al. (2021)), and manifold learning (Camastra et al. (2023)), showcasing their versatility and effectiveness. As a classical non-parametric Bayesian model (Williams & Rasmussen (2006)), GPs offer several advantages: ease of training, resistance to overfitting, uncertainty estimation, and the ability to incorporate prior knowledge. However, the significant time complexity associated with GPs limits their application, rendering them computationally intractable with large-scale datasets.

To enhance the computational efficiency of GPs, various methods have been developed, primarily focusing on sparse approximation techniques (Liu et al. (2020); Snelson & Ghahramani (2007)). These methods assume local conditional independence of labels within the training set and use inducing points to accelerate computation. This approach of summarizing the input space with strategically chosen inducing points has proven to be highly effective over years of research, leading to advancements such as SoR (Silverman (1985); Wahba et al. (1998)), DTC (Csató & Opper (2002); Seeger et al. (2003)), FITC (Snelson & Ghahramani (2005)), and PITC (Quinonero-Candela & Rasmussen (2005)).

The choice of inducing points is central to GP approximation. Initially, inducing points are selected from the training data (Smola & Bartlett (2000); Seeger et al. (2003)). However, Snelson & Ghahramani (2005) relaxes this constraint, proposing that inducing points can be viewed as auxiliary pseudo-inputs representing the spatial structure of the input data. A natural approach is to cluster the input data and assign each point to its nearest cluster center. In related studies, methods like farthest point clustering (Gonzalez (1985)) and random clustering (Sibuya (1993)) have been used to reduce computational burden. An innovative approach involves computing each cluster center using a separate Gaussian Process (Park & Choi (2010)). This method significantly improves computational efficiency by simplifying calculations with sparse matrices, although it may slightly increase prediction error when points are sparse within each partition.

While studying target-based partition strategies is equally important, research in this area remains limited. This approach is crucial for addressing challenges in data imbalance, which is a pervasive

issue in real-world data collection (Oommen et al. (2011); Spelmen & Porkodi (2018); Krawczyk (2016)). The concept of balancing data through reweighing the label space originated from studies on imbalanced categorical data (Huang et al. (2016); Fernández et al. (2011)). Recently, DIR (Yang et al. (2021)) has advanced this approach by applying kernel smoothing to continuous targets, achieving state-of-the-art performance on complex multi-model regression tasks. However, methods like LDS and FDS designed for data smoothing do not provide uncertainty assessment—an essential aspect for evaluating the performance of learning algorithms.

In this paper, we propose *Probabilistic Feature Smoothed Partially Independent Training Conditional Approximation* (PFS-PITC), a target-based partition and smoothing strategy to leverage the data approximation flexibility of PITC. Labels are divided into equidistant intervals to address data imbalance in training labels. Kernel smoothing is then applied, followed by a Gaussian sampling procedure to generate clustering centers for each label layer. According to theoretical analysis and multiple experiments, PFS-PITC brings reliable performance boost on datasets with underrepresented label space, providing a feasible way to adopt Gaussian Process to imbalanced regression missions.

## 2 RELATED WORK

### 2.1 SPARSE GAUSSIAN PROCESS APPROXIMATIONS

Gaussian Processes (GPs) are versatile probabilistic models that represent an underlying function as a distribution over possible functions. This framework allows for the integration of prior knowledge through prior mean and kernel functions, facilitating the accurate modeling of complex, non-linear relationships in real-world data (Marrel et al. (2008); Jones & Johnson (2009)). However, the application of GP is often limited by its unfavourable time scaling. The $O(N^3)$ cost of matrix inversion during training and the $O(N^2)$ cost per prediction limit the application of GPs to large-scale datasets. To address this drawback, several sparse GP approximations have been developed, reducing training time to $O(NM^2)$ and testing time to $O(M^2)$ Csató & Opper (2002); Snelson & Ghahramani (2005) (where $N$ and $M$ is the number of training and inducing samples).

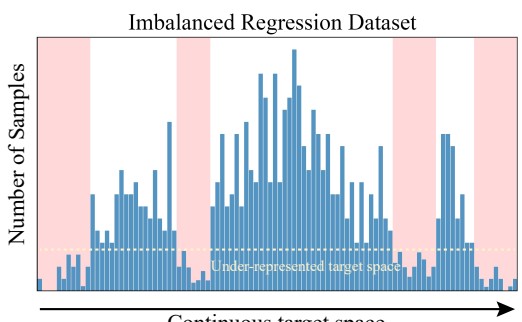

Figure 1: PFS-PITC aims to enhance the performance of Gaussian Processes on imbalanced regression datasets by minimizing the impact of under-represented label spaces.

Sparse GP approximations are derived from relaxing the conditional probability of the latent function $\mathbf{f}$ and the observation function $\mathbf{f}_T$ given the inducing variables, as comprehensively reviewed in Quinonero-Candela & Rasmussen (2005). To elucidate the distinctions among these closely related approximations, we will summarize the specific relaxations of several popular methods:

- The Deterministic Training Conditional Approximation (DTC)(Csató & Opper (2002); Seeger et al. (2003))

$$q_{\text{DTC}}(\boldsymbol{f}|\bar{\boldsymbol{f}}) = \mathcal{N}(\boldsymbol{f}; \boldsymbol{K}_{N,M}\boldsymbol{K}_M^{-1}\bar{\boldsymbol{f}}, \mathbf{0}), \tag{1}$$

- The Fully Independent Training Conditional Approximation (FITC)(Snelson & Ghahramani (2005))

$$q_{\text{FITC}}(\boldsymbol{f}|\bar{\boldsymbol{f}}) = \prod_i p(\boldsymbol{f}_i|\bar{\boldsymbol{f}}) = \mathcal{N}(\boldsymbol{f}; \boldsymbol{K}_{N,M}\boldsymbol{K}_M^{-1}\bar{\boldsymbol{f}}, \text{diag}(\boldsymbol{K}_N - \boldsymbol{Q}_N)), \tag{2}$$

- The Partially Independent Training Conditional Approximation (PITC)(Quinonero-Candela & Rasmussen (2005))

$$q_{\text{PITC}}(\boldsymbol{f}|\bar{\boldsymbol{f}}) = \prod_s p(\boldsymbol{f}_{B_s}|\bar{\boldsymbol{f}}) = \mathcal{N}(\boldsymbol{f}; \boldsymbol{K}_{N,M}\boldsymbol{K}_M^{-1}\bar{\boldsymbol{f}}, \text{blockdiag}(\boldsymbol{K}_N - \boldsymbol{Q}_N)), \tag{3}$$

where $\boldsymbol{Q}_{A,B} = \boldsymbol{K}_{A,M}\boldsymbol{K}_M^{-1}\boldsymbol{K}_{M,B}$, and $\boldsymbol{f}, \bar{\boldsymbol{f}}$ are the latent function vectors on training and inducing points.

Despite their similar assumptions about the conditional distribution of the latent function, these three approximations rest on different premises regarding dependencies. In DTC, the conditional distribution is treated as a point mass, indicating that the inducing points encapsulate all the necessary information for calculating latent functions. In contrast, FITC and PITC replace the deterministic relationship between $\bar{\mathbf{f}}$ and $\bar{\mathbf{f}}$ with Gaussian distributions, based on the assumptions of full and partial independence, respectively. PITC introduces group partitioning to divide the input points, assuming conditional independence of latent functions between groups. This approach balances computational efficiency with information loss by leveraging partial independence and allowing flexible partitioning of groups, thus integrating inducing points effectively.

Although these sparse GP approximations effectively reduce the computational burden of GP calculations, research has often overlooked their application to imbalanced regression problems. Among the mainstream GP approximations discussed, PITC is considered most suitable for leveraging uniformly distributed inducing points for two major reasons.

Firstly, unlike approximations that use point mass distributions, such as SoR and DTC, PITC allows for latent function uncertainty given the inducing functions. This flexibility avoids overly rigid relationships that can lead to shallow predictive variance and a higher risk of data overfitting.

Secondly, PITC assumes block-wise conditional probability independence, which is well suited for modeling elements within respective label bins. This characteristic can be discovered from the "PITC kernel function", given by $k^{\text{PITC}}(\boldsymbol{x}, \boldsymbol{x}') = \boldsymbol{Q}(\boldsymbol{x}, \boldsymbol{x}') + \mathbf{1}_{\text{x,x}' \in \text{B}_s}[k(\boldsymbol{x}, \boldsymbol{x}') - \boldsymbol{Q}(\boldsymbol{x}, \boldsymbol{x}')](\mathbf{1}_{(\cdot)}$ denotes the indicator function). This function balances the impact of group partitioning with a straightforward kernel distance, resulting in a more comprehensive quantification of distances between input variables.

## 2.2 IMBALANCED REGRESSION

Imbalanced regression is less explored compared to imbalanced classification problems (Zou et al. (2016); Feng et al. (2021)). Most existing regression methods for imbalanced data are variants of the SMOTE algorithm (Branco et al. (2017a); Torgo et al. (2013b)). These methods create artificial samples to oversample rare targets, either by interpolating training data (Torgo et al. (2013b); Rahim et al. (2019)) or by applying Gaussian noise augmentation (Branco et al. (2017a)). Despite their similar origins, regression-focused SMOTE algorithms share several limitations. Firstly, they fail to utilize the distance between continuous input labels effectively. The interpolation of inputs and labels relies on classification discreteness, leading to bias in the continuous feature space. Additionally, high-dimensional data pose significant challenges for oversampling algorithms, as synthetic data generated through linear interpolation often lack realism, which can further degrade model performance.

To leverage the statistical distribution uniformity of input features and labels, DIR (Yang et al. (2021)) proposes Feature Distribution Smoothing (FDS) and Label Distribution Smoothing (LDS), which apply kernel smoothing to latent features and labels, respectively. FDS and LDS partition features and labels into continuous bins, thereby addressing data imbalance and achieving state-of-the-art performance. Although DIR is compatible with various downstream networks, it primarily focuses on minimizing RMSE loss in regression and provides no uncertainty assessment.

## 3 BACKGROUND

### 3.1 NOTATIONS AND PROBLEM REVIEW

An imbalanced regression dataset is a dataset $\{(\boldsymbol{x}_i, y_i)\}_{i=1}^N$, where $N$ is the number of training points, $\boldsymbol{x}_i \in \mathbb{R}^d$ is the input, and $y_i \in \mathcal{Y} \subset \mathbb{R}$ is the corresponding label with imbalanced distribution in the continuous label space $\mathcal{Y}$. To measure the imbalance of label space within training dataset, we divide the label space $\mathcal{Y}$ into $B$ equidistant intervals with length $C$, i.e. $\mathcal{Y} = \bigcup_{b=1}^{B} \mathcal{Y}_b$, where $\mathcal{Y}_b = [y^{(b-1)}, y^{(b)}), |\mathcal{Y}_b| = C, b \in \mathbb{B} = \{1, \dots, B\} \subset \mathbb{Z}^+$.

## 3.2 GP WITH PITC APPROXIMATION

**Gaussian Process.** In the Gaussian Process framework, the latent function follows a multivariate Gaussian distribution (Williams & Rasmussen (2006); Kanagawa et al. (2018)). A typical Gaussian Process can be denoted as $\mathbf{f} \sim \mathcal{GP}(m(\cdot), k(\cdot, \cdot))$. Without specific prior knowledge, $m(\mathbf{x}) = 0$ is assumed by default. As for the kernel function, popular choices include the Linear kernel, the Spectral Mixture kernel, the Radial Basis Function (RBF) kernel, and the Cosine-Similarity kernel. According to the study of Reproducing Kernel Hilbert Spaces (RKHS) (Kanagawa et al. (2018)), a kernel function can also be viewed as an inner product defined in the feature space with a mapping $\varphi : X \to V, k(\boldsymbol{x}, \boldsymbol{x}') = \langle \varphi(\boldsymbol{x}), \varphi(\boldsymbol{x}') \rangle_V$. This observation enables the use of deep neural networks (DNNs) as feature extractors for specific needs (Wilson et al. (2016); Patacchiola et al. (2020); Yang et al. (2019)). The projection from input space to latent space does not require prior knowledge of the kernel function, making it a desirable approach to enhance the performance of GP models.

To specify the relationship between the observed data and the latent function, Gaussian Processes typically assume that the observation process introduces noise to the observed outputs, leading to the following regression problem:

$$y_i = f(\mathbf{x}_i) + \epsilon_i, \quad \epsilon_i \sim \mathcal{N}(0, \sigma^2). \tag{4}$$

**PITC approximation.** To approximate the Gaussian Process, the inducing point set and the inducing variable set $(\bar{\boldsymbol{X}}, \bar{\boldsymbol{f}}) = \{\bar{x}_m\}_{m=1}^M, \{\bar{f}_m\}_{m=1}^M, M \ll N$ are introduced to represent the distribution of input data. The GP prior of $\boldsymbol{f}, \boldsymbol{f}_T$ can now be approximated by:

$$p(\boldsymbol{f}, \boldsymbol{f}_T) \approx q(\boldsymbol{f}, \boldsymbol{f}_T) = \int q(\boldsymbol{f}_T | \bar{\boldsymbol{f}}) q(\boldsymbol{f} | \bar{\boldsymbol{f}}) p(\bar{\boldsymbol{f}}) \mathrm{d}\bar{\boldsymbol{f}} = \mathcal{N}(\boldsymbol{f}, \boldsymbol{f}_T; \boldsymbol{0}, \boldsymbol{K}_{N+T}^{\mathrm{PITC}}), \tag{5}$$

$$\boldsymbol{K}_{N+T}^{\mathrm{PITC}} = \begin{bmatrix} \boldsymbol{Q}_N + \mathrm{blockdiag}(\boldsymbol{K}_N - \boldsymbol{Q}_N) & \boldsymbol{Q}_{NT} \\ \boldsymbol{Q}_{TN} & \boldsymbol{K}_T \end{bmatrix}. \tag{6}$$

According to PITC, the probability distribution of predictive labels conditional on observed labels follows Gaussian distribution similar to classical GP: $p(\boldsymbol{y}_T | \boldsymbol{y}) = \mathcal{N}(\boldsymbol{y}_T; \boldsymbol{\mu}_T^{\mathrm{PITC}}, \boldsymbol{\Sigma}_T^{\mathrm{PITC}})$, where

$$\boldsymbol{\mu}_T^{\mathrm{PITC}} = \boldsymbol{Q}_{TN}[\boldsymbol{K}_N^{\mathrm{PITC}} + \sigma^2 \boldsymbol{I}]^{-1} y, \boldsymbol{\Sigma}_T^{\mathrm{PITC}} = \boldsymbol{K}_{TN} - \boldsymbol{Q}_{TN}[\boldsymbol{K}_N^{\mathrm{PITC}} + \sigma^2 \boldsymbol{I}]^{-1} \boldsymbol{Q}_{NT} + \sigma^2 \boldsymbol{I}. \tag{7}$$

**Remark.** It is important to note that the marginal probability distributions of training and test latent functions are not identical. Specifically, $q(\boldsymbol{f}) = \mathcal{N}(\boldsymbol{f}; \boldsymbol{0}, \boldsymbol{Q}_N + \mathrm{blockdiag}(\boldsymbol{K}_N - \boldsymbol{Q}_N))$ and $q(\boldsymbol{f}_T) = \mathcal{N}(\boldsymbol{f}_T; \boldsymbol{0}, \boldsymbol{K}_T)$. This indicates an assumption of prior knowledge regarding whether a data point belongs to the training or test set. While this characteristic denies the PITC approximation as an exact Gaussian Process, the assumption about data partitioning does not diminish its effectiveness for regression tasks.

# 4 PROBABILISTIC FEATURE SMOOTHED PARTIALLY INDEPENDENT TRAINING CONDITIONAL APPROXIMATION

## 4.1 KERNEL SMOOTHING OF STATISTIC FEATURES

To capture the non-linear structure among the training points, we first apply feature extraction using a simple neural network. Let $\mathcal{F}_{\boldsymbol{\theta}}$ be a feature extractor parameterized by $\boldsymbol{\theta}$. For an input point $\boldsymbol{x}_i$, its feature is denoted as $\boldsymbol{z}_i = \mathcal{F}_{\boldsymbol{\theta}}(\boldsymbol{x}_i)$. Let $(\boldsymbol{\mu}_b, \boldsymbol{\Sigma}_b)$ represent the mean and covariance of features $\{\boldsymbol{z}_i\}_{i=1}^{N_b}$ in the $b$-th bin($y_i \in \mathcal{Y}_b$). In practice, the mean and covariance are estimated empirically as follows:

$$\boldsymbol{\mu}_b = \hat{\mathbb{E}}[\boldsymbol{Z}_b] = \frac{1}{N_b} \sum_{i=1}^{N_b} \boldsymbol{z}_i, \boldsymbol{\Sigma}_b = \hat{\mathrm{Var}}[\boldsymbol{Z}_b] = \frac{1}{N_b - 1} \sum_{i=1}^{N_b} (\boldsymbol{z}_i - \boldsymbol{\mu})(\boldsymbol{z}_i - \boldsymbol{\mu})^\top. \tag{8}$$

The continuity of feature statistics across nearby bins was first observed in Yang et al. (2021). In the learned feature space of a regression task, the cosine similarity between feature means and variances decreases monotonically with increasing label distance. This observation led the authors

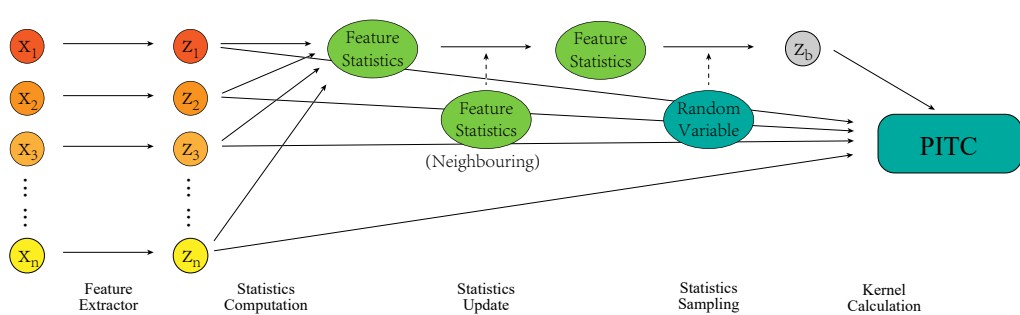

Figure 2: General architecture of PFS-PITC for the b-th bin. In the beginning, feature extractor embed input data into latent space. Next, feature statistics is computed to create approximate distribution for features. In the following process, random variable is used to sample from the approximate distribution. Eventually, with inducing points and input points ready, kernel matrix is computed for PITC training and prediction.

to propose Feature Distribution Smoothing (FDS) as a momentum update calibration layer following feature extraction. The FDS operation adjusts each feature based on neighboring statistics, partially compensating for sampling imbalances in the raw input space.

The kernel calibration of statistics in our method is implemented as follows. We impose kernel smoothing on $(\boldsymbol{\mu}_b, \boldsymbol{\Sigma}_b)$ with a kernel function $k_{\boldsymbol{\psi}}(\cdot, \cdot)$ parameterized with hyperparameters $\boldsymbol{\psi}$. The smoothed statistics are given as follows:

$$\tilde{\boldsymbol{\mu}}_b = \sum_{b' \in \mathbb{B}} k_{\boldsymbol{\psi}}(y_b, y_{b'}) \boldsymbol{\mu}_{b'}, \tilde{\boldsymbol{\Sigma}}_b = \sum_{b' \in \mathbb{B}} k_{\boldsymbol{\psi}}(y_b, y_{b'}) \boldsymbol{\Sigma}_{b'}. \tag{9}$$

While the calibrated feature distribution better represents the input features, it still does not meet the requirements of GP, which does not accept probability distributions as input. Therefore, sampling is necessary to generate inducing inputs for PITC approximation. Assuming that $\{\boldsymbol{z}_i\}_{i=1}^{N_b}$ follows a Gaussian distribution as a prior, we approximate the smoothed distribution as $q(\boldsymbol{z}_i) = \mathcal{N}(\boldsymbol{z}_i; \boldsymbol{\mu}_b, \boldsymbol{\Sigma}_b)$. This allows for sampling from a Gaussian distribution:

$$\bar{\boldsymbol{Z}} = \{\boldsymbol{\mu}_b + \eta \boldsymbol{\Sigma}_b\}_{b \in \mathbb{B}}, \eta \sim \mathcal{N}(\boldsymbol{0}, \boldsymbol{I}). \tag{10}$$

## 4.2 OPTIMIZATION OBJECTIVE

To model the heterogeneous feature distribution within each bin, we introduce bin-wise observation noise. The observed label is given by $y_i = f(\boldsymbol{z}_i) + \epsilon_i, \epsilon_i \sim \mathcal{N}(0, \sigma_b^2), y_i \in \mathcal{Y}_b$. The conditional probability distribution of $\boldsymbol{y}$ given $\boldsymbol{f}$ becomes:

$$p(\boldsymbol{y}|\boldsymbol{f}) = \mathcal{N}(\boldsymbol{y}; \boldsymbol{f}, \boldsymbol{\Sigma}), \boldsymbol{\Sigma} = \mathrm{diag}(\sigma_1^2, \dots, \sigma_B^2). \tag{11}$$

In Gaussian Process (GP) optimization, the log marginal likelihood (NLL) is commonly used as the objective function to minimize. Following a similar derivation for PITC, the expression for the NLL is given by:

$$\mathcal{L}^{\mathrm{PITC}} = \frac{1}{2} \log |\boldsymbol{K}_N^{\mathrm{PITC}} + \boldsymbol{\Sigma}| + \frac{1}{2} \boldsymbol{y}^{\top} (\boldsymbol{K}_N^{\mathrm{PITC}} + \boldsymbol{\Sigma})^{-1} \boldsymbol{y} + \frac{N}{2} \log(2\pi), \tag{12}$$

where $\boldsymbol{K}_N^{\mathrm{PITC}} = \boldsymbol{Q}_N + \mathrm{blockdiag}(\boldsymbol{K}_N - \boldsymbol{Q}_N), (\boldsymbol{K}_N)_{i,j} = k_{\boldsymbol{\phi}}(\mathcal{F}_{\boldsymbol{\theta}}(\boldsymbol{x}_i), \mathcal{F}_{\boldsymbol{\theta}}(\boldsymbol{x}_j)), (\boldsymbol{Q}_N)_{i,j} = \boldsymbol{K}_{\boldsymbol{x}_i, M} \boldsymbol{K}_M^{-1} \boldsymbol{K}_{M, \boldsymbol{x}_j}$.

The overall procedure of conducting PFS-PITC on a regression mission is provided in Algorithm1.

## 4.3 THEORETICAL ANALYSIS

In this section, we manage to conduct theoretical analysis on stability of domain generalization of PFS-PITC. We deduce the tail bound for PFS estimator and generalization bound of PFS estimator for finite observation space.

---

**Algorithm 1** Training and test procedure of PFS-PITC

---

**Require:** Train dataset $\mathcal{D} = \{(\boldsymbol{x}_i, y_i)\}_{i=1}^N$, test dataset $\mathcal{D}_{\text{test}} = \{(\boldsymbol{x}_i, y_i)\}_{i=1}^T$.
**Parameters:** Feature extractor parameters $\boldsymbol{\theta}$, kernel parameters $\boldsymbol{\phi}$.
**Hyperparameters:** Bin index $\mathbb{B}$, Kernel smoothing parameters $\boldsymbol{\psi}$, learning rate $\alpha, \beta$, update rate $\gamma$.

1: **function** TRAIN($\mathcal{D}, \boldsymbol{\theta}, \boldsymbol{\phi}, \boldsymbol{\psi}, \alpha, \beta$)
2:   **while** not done **do**
3:     Sample batch $\mathcal{T} = (\boldsymbol{X}, \boldsymbol{y}) \sim \mathcal{D}$
4:     Extract Feature $\boldsymbol{Z} = \mathcal{F}_{\boldsymbol{\theta}}(\boldsymbol{X})$
5:     **for** $b \in \mathbb{B}$ **do**
6:       Compute statistical features $(\boldsymbol{\mu}_b, \boldsymbol{\Sigma}_b)$                   $\triangleright$ See Equation(8)
7:       Computed smoothed statistical features$(\tilde{\boldsymbol{\mu}}_b, \tilde{\boldsymbol{\Sigma}}_b)$     $\triangleright$ See Equation(9)
8:       Implement update $(\boldsymbol{\mu}_b, \boldsymbol{\Sigma}_b) \leftarrow (1-\gamma) * (\boldsymbol{\mu}_b, \boldsymbol{\Sigma}_b) + \gamma * (\tilde{\boldsymbol{\mu}}_b, \tilde{\boldsymbol{\Sigma}}_b)$
9:     **end for**
10:    Sample inducing points $\bar{\boldsymbol{Z}}$                          $\triangleright$ See Equateion(10)
11:    Compute NLL $\mathcal{L}^{\text{PITC}}$                            $\triangleright$ See Equateion(12)
12:    Update parameters $\boldsymbol{\theta} \leftarrow \boldsymbol{\theta} - \alpha \nabla_{\boldsymbol{\theta}} \mathcal{L}^{\text{PITC}}, \boldsymbol{\phi} \leftarrow \boldsymbol{\phi} - \beta \nabla_{\boldsymbol{\phi}} \mathcal{L}^{\text{PITC}}$
13:   **end while**
14:   **return** $\boldsymbol{\theta}, \boldsymbol{\phi}, \bar{\boldsymbol{Z}}$
15: **end function**

16: **function** TEST($\mathcal{D}_{\text{test}}, \boldsymbol{\theta}, \boldsymbol{\phi}$)
17: Sample batch $\mathcal{T} = (\boldsymbol{X}_T, \boldsymbol{y}_T) \sim \mathcal{D}_{\text{test}}$
18: Extract Feature $\boldsymbol{Z}_T = \mathcal{F}_{\boldsymbol{\theta}}(\boldsymbol{X}_T)$
19: **return** $p(\boldsymbol{y}_T | \boldsymbol{Z}_T, \bar{\boldsymbol{Z}}, \boldsymbol{Z}, \boldsymbol{y})$                        $\triangleright$ See Equateion(7)
20: **end function**

---

**Notation.** We start from a balanced regression dataset $\{(\boldsymbol{x}_i, y_i)\}_{i=1}^N$ and a equidistant partition of observation space $\mathcal{Y} = \bigcup_{b=1}^{|\mathbb{B}|} \mathcal{Y}_b \subset \mathbb{R}, |\mathcal{Y}_b| = C$. To model the imbalanced sampling, we introduce binary revealing set $\mathbb{O} = \{0, 1\}^N$ to label whether $(\boldsymbol{x}_i, y_i)$ is sampled. In addition, probability set $\boldsymbol{P} = \{P_b\}_{b=1}^{|\mathbb{B}|}, P_b = \mathbb{P}(y_i \in \mathcal{Y}_b)$ describes the marginal probability distribution for each bin. For each bin $b$, index indicator set $\mathbb{U}_b = \{i | y_i \in \mathcal{Y}_b\}$ denotes the index of samples, and $\mathbb{S}_b = \{i | O_i = 1, y_i \in \mathcal{Y}_b\}$ denotes index given the imbalanced observation made on raw dataset.

**Lemma 1.** *(Tail bound for PFS Estimator). For any given $\hat{y}$ and $y$, with probability $1 - \eta$, the PFS estimator $\hat{R}_{PFS}(\hat{y}|\tilde{P})$ does not deviate from its expectation $\mathbb{E}_O[\hat{R}_{PFS}(\hat{y}|\tilde{P})]$ by more than:*

$$|\hat{R}_{PFS}(\hat{y}|\tilde{P}) - \mathbb{E}_O[\hat{R}_{PFS}(\hat{y}|\tilde{P})]| \leq \frac{\Delta}{|\mathbb{B}|} \sqrt{\frac{\log(2|\mathcal{H}|/\eta)}{2}} \sqrt{\sum_{b=1}^{|\mathbb{B}|} \frac{1}{\tilde{P}_b^2}}. \tag{13}$$

**Theorem 1.** *(Propensity-Scored ERM Generalization Error Bound of PFS). In imbalanced regression with bins partition $\mathbb{B}$, for any finite hypothesis space of predictions $\mathcal{H} = \{\hat{y}_1, \ldots \hat{y}_{|\mathcal{H}|}\}$, the transductive prediction error of the empirical risk minimizer $\hat{y}^{ERM}$, using the PFS estimator with estimated propensities $\tilde{P}(\tilde{P}_b > 0)$ and given training observations $O$ from $\mathcal{Y}$ with independent Bernoulli propensities P, is bounded by:*

$$R(\hat{y}^{ERM}) \leq \hat{R}_{PFS}(\hat{y}^{ERM}|\tilde{P}) + \underbrace{\frac{\Delta}{|\mathbb{B}|} \sum_{b=1}^{|\mathbb{B}|} |1 - \frac{P_b}{\tilde{P}_b}|}_{Bias} + \underbrace{\frac{\Delta}{|\mathbb{B}|} \sqrt{\frac{\log(2|\mathcal{H}|/\eta)}{2}} \sqrt{\sum_{b=1}^{|\mathbb{B}|} \frac{1}{\tilde{P}_b^2}}}_{Variance}. \tag{14}$$

**Remark.** This theorem establishes an upper bound on the true risk as estimated by the PFS estimator. Without probabilistic feature smoothing, substituting the smoothed $\tilde{P}_b$ with the observed $P_b$ results in a bias of 0 but significantly increases the variance term due to $P_b \approx 0$ for minority bins. Conversely, probabilistic feature smoothing aims to smooth each probabilistic feature $P_b$ with its neighboring features, leading to an estimator that more closely aligns with the true risk estimator.

Following the feature smoothing operation, the probabilistic estimation for minority bins ($P_b \approx 0$) is enhanced, significantly reducing the variance term, albeit at the cost of an increase in the bias term. In the context of imbalanced regression, balanced probabilistic features minimize the risk contribution from the under-sampled label subspace, resulting in a substantially lower generalization error.

For specific definitions of related concepts, we refer readers to the appendix for more information.

## 5 EXPERIMENTS

### 5.1 SYNTHETIC DATA

We begin by comparing PFS-PITC, local GP, and GP on a simple synthetic regression dataset. This experiment is designed to emphasize the performance differences among these GP-related regression algorithms in a variable space with sparse training samples. The ground-truth function is defined with observation noise as follows: $f(x) = \exp(x) * \cos(2\pi x) * (2 + \epsilon), \epsilon \sim \mathcal{N}(0, 1)$. Training inputs are gathered from the combination of two separate distributions on input variables: $\mathcal{D} = \{(x_i, y_i)\} \cup \{(x'_j, y'_j)\}, x_i \sim \mathcal{N}(0, 1), x'_j \sim \mathcal{N}(2, 1)$. Test inputs are gathered from a uniform distribution without observation noise:$\mathcal{D}_{\text{test}} = \{(x_i^*, y_i^*)\}, x_i^* \sim U(2.5, 3.5), \epsilon = 0$. This artificial experiment presents two major challenges for regression models: 1) The ground-truth function incorporates periodic patterns that vary in amplitude, mimicking the scaling instability of labels in real-world data. 2) The observation space in the training and test sets alternates in intervals, simulating data sampling imbalance.

We train all three methods using the RBF kernel and identical hyperparameters to ensure comparable experimental outcomes. The test bias of PFS-PITC achieves an MSE of 26.290 and an MAE of 4.305, significantly outperforming GP (MSE: 75.667, MAE: 6.824) and local GP (MSE: 82.036, MAE: 7.364). Local GP establishes 4 separate GPs for the clusters identified by KMeans, resulting in a prediction output similar to that of vanilla GP in most regions. The difference in fitting accuracy is illustrated in Figure 3, where the prediction mean of PFS-PITC is closer to the test points, despite the training points being noticeably scattered(at x=2.5 and x=3.0). These findings support our assertion that PFS-PITC outperforms both vanilla GP and local GP in addressing imbalanced samples and scaling instability.

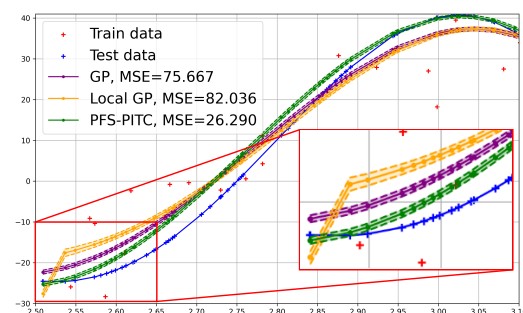

Figure 3: Comparison of synthetic experiments among PFS-PITC, GP, and local GP. PFS-PITC more effectively approximates the ground-truth function, particularly in scenarios with varying amplitude and sparse training inputs around the test interval.

### 5.2 REGRESSION DATASETS

In this section, we evaluate the performance of PFS-PITC on two real-world regression datasets: *Combined Cycle Power Plant* and *Concrete Compressive Strength*. Multiple approaches for imbalanced regression are implemented in conjunction with Gaussian Process for performance comparison. Each algorithm is tested five times to provide mean and bias values for stability analysis. The evaluation metrics for this experiment include Mean Squared Error (MSE), Mean Absolute Error (MAE), and Negative Log-Likelihood (NLL).

**Combined Cycle Power Plant.** The *Combined Cycle Power Plant* (CCPP) dataset, created by Pnar Tfekci (Tüfekci (2014)) and Heysem Kaya (Zhao & Kok Foong (2022)), comprises 9,568 data points collected from a Combined Cycle Power Plant operating at full capacity over six years (2006-2011). The regression task involves predicting the net hourly electrical energy output (EP) of the plant based on four key input features: hourly average ambient temperature (T), ambient pressure (AP), relative humidity (RH), and exhaust vacuum (V).

**Concrete Compressive Strength.** The *Concrete Compressive Strength*(CCS) dataset (Yeh (1998)) formulates the compressive strength of concrete as a regression problem. It includes eight features:

the density (measured in $kg/m^3$) of cement, blast furnace slag, fly ash, water, superplasticizer, coarse aggregate and fine aggregate, and the age (measured in days) since cement manufacturing. Regression models aim to learn the concrete compressive strength (measured in MPa) from 1,030 instances.

| Dataset | Combined Cycle Power Plant | | |
|---|---|---|---|
| Methods | MSE↓ | MAE↓ | NLL↓ |
| VANILLA GP | 346.247±101.425 | 14.789±2.834 | 1047.741±15.246 |
| **SMOGN**(Branco et al. (2017b)) | 304.526±111.192 | 14.230±2.524 | 1228.306±1830.890 |
| **SMOTER**(Torgo et al. (2013a)) | 170.877±70.148 | 10.595±2.443 | 4115.621±273.998 |
| **Random Undersampling** | 254.779±129.659 | 12.655±3.986 | 1936.639±14.778 |
| **Gaussian Noise**(Branco et al. (2019)) | 238.848±84.355 | 12.351±2.709 | 1079.346±4.427 |
| **CNN**(Hart (1968)) | 339.045±102.014 | 15.628±2.033 | 4712.031±551.982 |
| **FDS**(Yang et al. (2021)) | 398.846±109.765 | 15.738±2.577 | **1038.242±32.017** |
| **LDS**(Yang et al. (2021)) | 470.120±76.602 | 17.052±1.882 | 1437.778±219.296 |
| **PFS-PITC**(Bin num=80) | 123.332±4.147 | 8.584±0.197 | 954.572±3.966 |
| **PFS-PITC**(Bin num=90) | 121.136±4.821 | 8.524±0.105 | 955.641±5.031 |
| **PFS-PITC**(Bin num=100) | **120.650±5.831** | **8.510±0.081** | 955.893±4.690 |
| **PFS-PITC (BEST)** VS. VANILLA GP | +225.597 | +6.279 | +93.169 |

Table 1: Imbalanced regression on Combined Cycle Power Plant dataset

| Dataset | Concrete Compressive Strength | | |
|---|---|---|---|
| Methods | MSE↓ | MAE↓ | NLL↓ |
| VANILLA GP | 127.956±13.79 | 7.903±0.156 | 954.067±36.792 |
| **SMOGN**(Branco et al. (2017b)) | 125.425±16.683 | 8.391±0.856 | 707.862±94.103 |
| **SMOTER**(Torgo et al. (2013a)) | 119.224±20.550 | 8.783±0.488 | 1736.234±107.566 |
| **Random Undersampling** | 217.431±87.941 | 10.564±2.061 | **498.286±113.213** |
| **Gaussian Noise**(Branco et al. (2019)) | 140.873±19.641 | 8.522±0.509 | 1222.020±124.957 |
| **CNN**(Hart (1968)) | 121.949±11.501 | 7.150±0.666 | 4286.112±250.309 |
| **FDS**(Yang et al. (2021)) | 178.571±50.282 | 9.577±1.285 | 1108.508±89.248 |
| **LDS**(Yang et al. (2021)) | 106.176±10.289 | 8.133±0.541 | 1747.272±739.201 |
| **PFS-PITC**(Bin num=80) | **102.863±4.725** | **6.929±0.213** | 910.858±109.423 |
| **PFS-PITC**(Bin num=90) | 104.678±5.616 | 6.968±0.188 | 905.848±111.521 |
| **PFS-PITC**(Bin num=100) | 104.494±5.172 | 6.964±0.201 | 870.887±54.999 |
| **PFS-PITC (BEST)** VS. VANILLA GP | +25.093 | +0.974 | +83.180 |

Table 2: Imbalanced regression on Concrete Compressive Strength dataset

During the preprocessing procedure, we eliminate duplicate observations of input variables and apply min-max normalization to mitigate potential distribution shifts. Subsequently, we divide the sample targets and corresponding variables into 100 equidistant intervals to assess target distribution. Instances within each bin are allocated to the training, validation, and test datasets in sequence until the quota is met, simulating the imbalance in training sample sampling and ensuring relatively uniform prediction demands.

In each dataset, we compare PFS-PITC with several common algorithms for imbalanced learning, and vanilla GP is used as the baseline. Smogn and Smoter, derived from SMOTE, are designed to interpolate instances to alleviate the sampling imbalance of rare targets in the training set. Random Undersampling, Gaussian Noise, and Condensed Nearest Neighbor provide alternative approaches by undersampling, injecting noise, and removing redundant samples, respectively. DIR (Yang et al. (2021)), along with LDS and FDS, proposes alternatives that leverage the continuity of label space in regression tasks to enhance performance. Despite their widespread use for addressing imbalanced datasets, some of these algorithms are incompatible with the training process of Gaussian Processes, which can lead to degraded prediction performance.

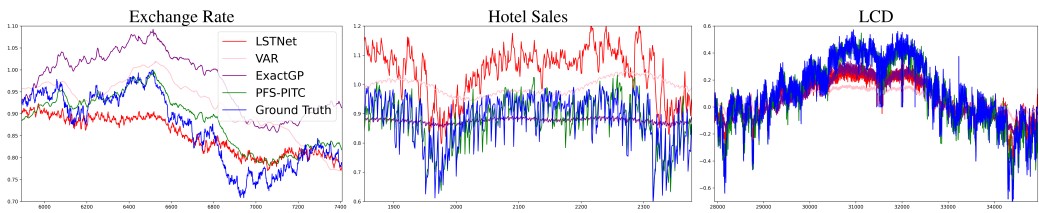

Figure 4: Prediction figure on Exchange-Rate, Hotel-Sales, and LCD dataset.

Quantitative analysis of this experiment is presented in Tables 1 and 2. Regarding NLL, most imbalanced regression approaches significantly alter the means, highlighting the varied effects of synthetic instances on the training process. Experiments with PFS-PITC are conducted with increasing partition numbers of bins to assess their impact. The outcome suggests that prediction bias remains stable regardless of the granularity of the partition. For MSE and MAE, PFS-PITC minimizes prediction loss, achieving (MSE: 120.650, MAE: 8.510) on the *CCPP* dataset and (MSE: 102.863, MAE: 6.929) on the *CCS* dataset.

The results reveal that PFS-PITC is competitive with most imbalanced regression approaches, regardless of prior knowledge about the label density distribution of the datasets. More importantly, PFS-PITC achieves the lowest MSE and MAE on both datasets, further validating its effectiveness in addressing imbalanced regression tasks.

### 5.3 TIME SERIES PREDICTION

To further evaluate the performance of PFS-PITC on diverse real-world datasets, we implement it on three time series datasets: the *Exchange Rate*, *Hotel Sales*, and *Local Climatological Data*. Mean Absolute Error (MAE) and Correlation Coefficient (CORR) are employed as evaluation metrics for this experiment.

**Exchange-Rate.** (Lai et al. (2018)). The exchange rate of a currency is assessed by the exchange ratio between the currency and the US Dollar per unit. *Exchange-Rate* dataset includes daily exchange rates for eight foreign countries—Australia, Britain, Canada, Switzerland, China, Japan, New Zealand, and Singapore—ranging from 1990 to 2016, comprising 7,588 data points. This dataset exhibits few long-term patterns, characterized by a prevalence of highly uncorrelated repetitive signals. We select Singapore as the target for prediction, using the other exchange rates as input features.

**Hotel-Sales.** (STR (2021)). The *Hotel-Sales* dataset contains data on hotel demand and revenue across eight major tourist destinations in the US. This dataset includes sales, daily occupancy, demand, and revenue for upper-middle-class hotels, aimed at estimating the impact of the COVID-19 pandemic on the tourism economy. We focus on the time series data for New York from 2013 to 2019, with 2,624 instances available for models to predict hotel occupancy based on past revenue and demand.

**Local Climatological Data.** (NOAA (2024)). *Local Climatological Data*(LCD), provided by NOAA, includes climatological data for nearly 1,600 U.S. locations over four years (2010-2013). This dataset comprises hourly weather features, with 11 meteorological attributes and Wet Bulb Celsius as the target. We focus on a subset of data from a single weather station over a specific month, consisting of 35,064 entries, to validate our model.

We select three common time series predictors for performance comparison: GP, VAR, and LSTNet. GP and VAR are chosen to establish a performance baseline for statistical methods. In contrast, LSTNet (Lai et al. (2018)) represents the capabilities of deep neural network (DNN) methods. This approach leverages a combination of CNNs, RNNs, and attention layers to extract short-term patterns among variables while also capturing long-term trends, resulting in a significant performance boost on complex real-world datasets.

During the preprocessing procedure, we apply maximum normalization to ensure numerical stability. The dataset is then split into training (60%), validation (20%), and test (20%) sets in sequence. To enhance information absorption, we employ a CNN feature extractor for feature extraction. For GP and PFS-PITC, we utilize a mixture of RBF, linear, and spectral mixture kernels to model the

| Dataset | | Exchange-Rate | | | Hotel-Sales | | | LCD | | |
|---|---|---|---|---|---|---|---|---|---|---|
| | | Horizon(Window=128) | | | Horizon(Window=220) | | | Horizon(Window=100) | | |
| Method | Metric | 3 | 6 | 12 | 3 | 6 | 12 | 3 | 6 | 12 |
| **LSTNet** | MAE | 0.040 | 0.037 | **0.038** | 0.053 | 0.068 | 0.069 | 0.053 | 0.068 | **0.049** |
| | CORR | 0.932 | 0.923 | **0.925** | 0.864 | 0.844 | 0.821 | 0.921 | 0.833 | 0.927 |
| **VAR** | MAE | 0.066 | 0.064 | 0.060 | 0.047 | 0.049 | 0.066 | 0.067 | 0.077 | 0.081 |
| | CORR | 0.827 | 0.886 | 0.765 | 0.846 | 0.842 | 0.798 | 0.812 | 0.805 | 0.782 |
| **GP** | MAE | 0.110 | 0.069 | 0.074 | 0.045 | 0.049 | 0.046 | 0.066 | 0.068 | 0.064 |
| | CORR | 0.864 | 0.605 | 0.679 | 0.726 | 0.699 | 0.755 | 0.869 | 0.860 | 0.839 |
| **PFS-PITC**(Bin num=30) | MAE | 0.030 | 0.060 | 0.053 | 0.039 | 0.036 | **0.035** | 0.036 | 0.050 | 0.061 |
| | CORR | **0.949** | 0.832 | 0.687 | 0.958 | 0.933 | **0.927** | 0.938 | 0.959 | 0.940 |
| **PFS-PITC**(Bin num=50) | MAE | **0.029** | 0.073 | 0.059 | **0.037** | **0.034** | 0.039 | 0.035 | 0.050 | 0.060 |
| | CORR | 0.939 | 0.841 | 0.837 | **0.981** | 0.912 | 0.934 | **0.957** | 0.961 | **0.942** |
| **PFS-PITC**(Bin num=100) | MAE | 0.036 | **0.034** | 0.069 | 0.038 | 0.038 | 0.036 | **0.034** | **0.047** | 0.062 |
| | CORR | 0.885 | **0.928** | 0.857 | 0.921 | **0.963** | 0.933 | 0.928 | **0.961** | 0.936 |

Table 3: Time Series Experiments on Exchange-Rate, Hotel-Sales, and LCD dataset.

heterogeneous patterns in the dataset. For further details on this experiment, we recommend that readers refer to the appendix.

The evaluation results of all four methods across the three datasets are presented in Table 3. The prediction horizon and retrospective window indicate the number of timestamps ahead of and behind the current time, respectively, and these parameters are varied to assess the stability of each algorithm. From the table, we observe that LSTNet demonstrates its ability to make accurate long-term predictions, minimizing prediction bias at longer horizons (horizon = 12) with results of (MAE: 0.038, CORR: 0.925) on the *Exchange-Rate* dataset and (MAE: 0.049, CORR: 0.927) on the *LCD* dataset. However, PFS-PITC delivers superior results for shorter horizons, while GP does not show significant superiority compared to its competitors. On the *Exchange-Rate* dataset, the best MAE of PFS-PITC improves from (0.110, 0.069, 0.074) by GP to (0.029, 0.034, 0.053), yielding an average performance boost of over 40%.

## 6 CONCLUSIONS

In this work, we introduce PFS-PITC, a target-based partition and smoothing strategy for GP approximation, aimed at enhancing the performance of Gaussian Processes on imbalanced regression tasks. PFS-PITC extends the classical PITC approximation by employing label bins to achieve a more balanced integration of variable information. Additionally, kernel smoothing is applied to reduce distribution discrepancies in the latent features within each bin. Compared to incorporating resampling techniques directly into GP, PFS-PITC offers a more effective solution for addressing imbalanced regression in the Gaussian Process Regression framework. Extensive empirical experiments demonstrate significant performance improvements with our approach.

### 6.1 BROADER IMPACT

Gaussian Processes (GPs) are widely employed for various data analysis tasks Li et al. (2019); Dutordoir et al. (2018). While their performances can be affected by sampling biases, such as domain shift or imbalanced sampling, using inducing points for approximation enables targeted countermeasures. PFS-PITC, inspired by the traditional PITC approximation, specifically addresses the issue of imbalanced sample distribution in the label space. Our method offers an effective solution for imbalanced regression, providing uncertainty estimation and enhanced inference explainability. Furthermore, it integrates seamlessly into the Gaussian Process Regression framework, requiring minimal adjustments and facilitating performance improvements on imbalanced datasets through a straightforward substitution of GP with our approach. This capability significantly enhances the applicability of GPs in various fields, including finance, healthcare, and environmental modeling, ultimately contributing to more robust data analysis in these critical areas.

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
