## A  THEORETICAL ANALYSIS

**Notation.** We start from a balanced regression dataset $\{(\boldsymbol{x}_i, y_i)\}_{i=1}^N$ and a equidistant partition of observation space $\mathcal{Y} = \bigcup_{b=1}^{|\mathbb{B}|} \mathcal{Y}_b \subset \mathbb{R}, |\mathcal{Y}_b| = C$. To model the imbalanced sampling, we introduce binary revealing set $\mathbb{O} = \{0, 1\}^N$ to label whether $(\boldsymbol{x}_i, y_i)$ is sampled. In addition, probability set $\boldsymbol{P} = \{P_b\}_{b=1}^{|\mathbb{B}|}, P_b = \mathbb{P}(y_i \in \mathcal{Y}_b)$ describes the marginal probability distribution for each bin. For each bin $b$, index indicator set $\mathbb{U}_b = \{i | y_i \in \mathcal{Y}_b\}$ denotes the index of samples, and $\mathbb{S}_b = \{i | O_i = 1, y_i \in \mathcal{Y}_b\}$ denotes index given the imbalanced observation made on raw dataset.

**Definition 1.** *The expectation over the revealing indicator is defined as:*

$$\mathbb{E}_O[\cdot] = \mathbb{E}_{O_i \sim \mathbb{P}(O_i = 1)}[\cdot].$$

**Definition 2.** *(True Risk). The true risk for estimating $\hat{y}$ is defined as:*

$$R(\hat{y}) = \frac{1}{|\mathbb{B}|} \sum_{b=1}^{|\mathbb{B}|} \frac{1}{|\mathbb{U}_b|} \sum_{i \in \mathbb{U}_b} \delta_i(y, \hat{y}),$$

*where $\delta_i(y, \hat{y})$ denotes any loss function with upper bound $0 \le \delta_i(y, \hat{y}) \le \Delta$.*

The definition of true risk is for the purpose of evaluation of the risk of observation space with no sampling bias.

**Definition 3.** *(Naive Estimator). The naive estimator for estimating $\hat{y}$ is defined as:*

$$\hat{R}_{NAIVE}(\hat{y}) = \frac{1}{\sum\limits_{b=1}^{|\mathbb{B}|} |S_b|} \sum_{b=1}^{|\mathbb{B}|} \sum_{i \in \mathbb{U}_b} \delta_i(y, \hat{y}).$$

The definition of naive estimator is for the purpose of evaluation of imbalanced sampled dataset.

**Definition 4.** *(IPS Estimator). The IPS estimator for estimating $\hat{y}$ is defined as:*

$$\hat{R}_{IPS}(\hat{y}) = \frac{1}{|\mathbb{B}|} \sum_{b=1}^{|\mathbb{B}|} \frac{1}{|\mathbb{U}_b|} \sum_{i \in \mathbb{U}_b} \frac{\delta_i(y, \hat{y})}{P_b}.$$

The definition of IPS estimator is originated from Schnabel et al. (2016), for the purpose of evaluation of imbalanced sampled dataset with corrected sampling probabilities.

**Definition 5.** *(PFS Estimator). The PFS estimator for estimating $\hat{y}$ is defined as:*

$$\hat{R}_{PFS}(\hat{y}) = \frac{1}{|\mathbb{B}|} \sum_{b=1}^{|\mathbb{B}|} \frac{1}{|\mathbb{U}_b|} \sum_{i \in \mathbb{U}_b} \frac{\delta_i(y, \hat{y})}{\tilde{P}_b},$$

*where $\{\tilde{P}_b\}_{b=1}^B$ represents the smoothed label distribution utilized in our PFS's objective function.*

The definition of PFS estimator is for the purpose of evaluation of imbalanced sampled dataset with smoothed probabilistic features.

**Definition 6.** *(Propensity-Scored ERM). The Propensity-Scored ERM for observation space $\mathcal{H}$ is defines as:*

$$\hat{y}^{ERM} = \arg\min_{\hat{y} \in \mathcal{H}} \{\hat{R}_{PFS}(\hat{y} | \tilde{P})\}.$$

**Lemma 1.** *(Tail bound for PFS Estimator). For any given $\hat{y}$ and $y$, with probability $1 - \eta$, the PFS estimator $\hat{R}_{PFS}(\hat{y} | \tilde{P})$ does not deviate from its expectation $\mathbb{E}_O[\hat{R}_{PFS}(\hat{y} | \tilde{P})]$ by more than:*

$$|\hat{R}_{PFS}(\hat{y} | \tilde{P}) - \mathbb{E}_O[\hat{R}_{PFS}(\hat{y} | \tilde{P})]| \le \frac{\Delta}{|\mathbb{B}|} \sqrt{\frac{\log(2|\mathcal{H}|/\eta)}{2}} \sqrt{\sum_{b=1}^{|\mathbb{B}|} \frac{1}{\tilde{P}_b^2}}. \tag{15}$$

*Proof.* According to Hoeffding's inequality states that for independent bounded random variables $Z_1, \ldots, Z_n$ that take values in intervals of sizes $\rho_1, \ldots, \rho_n$ with probability 1 and for any $\epsilon > 0$

$$\mathbb{P}(|\sum_k Z_k - \mathbb{E}[\sum_k Z_k]| \geq \epsilon) \leq 2\exp(\frac{-2\epsilon^2}{\sum_k \rho_k^2})$$

Defining $Z_k = \hat{R}_{PFS}(\hat{Y}^{\text{ERM}}|\tilde{P})$, $\mathbb{P}(Z_k = \frac{\delta_i(y,\hat{y})}{\tilde{P}_b}) = \tilde{P}_b$ and $\mathbb{P}(Z_k = 0) = 1 - \tilde{P}_b$. With $\epsilon_0 = |\mathbb{B}|\epsilon$, we have:

$$\mathbb{P}(||\mathbb{B}|\hat{R}_{\text{PFS}}(\hat{y}^{\text{ERM}}|\tilde{P}) - |\mathbb{B}|\mathbb{E}_O[\hat{R}_{\text{PFS}}(\hat{y}^{\text{ERM}}|\tilde{P})]| \geq \epsilon_0) \leq 2\exp(-\frac{2\epsilon_0^2}{\Delta^2 \sum_{b=1}^{|\mathbb{B}|} \frac{1}{|\mathbb{U}_b|} \sum_{i \in \mathbb{U}_b} \frac{1}{\tilde{P}_b^2}})$$

$$\Leftrightarrow \mathbb{P}(|\hat{R}_{\text{PFS}}(\hat{y}^{\text{ERM}}|\tilde{P}) - \mathbb{E}_O[\hat{R}_{\text{PFS}}(\hat{y}^{\text{ERM}}|\tilde{P})]| \geq \epsilon) \leq 2\exp(-\frac{2\epsilon^2|\mathbb{B}|^2}{\Delta^2 \sum_{b=1}^{|\mathbb{B}|} \frac{1}{\tilde{P}_b^2}})$$

Solving for $\epsilon$ completes the proof. $\square$

**Theorem 1.** *(Propensity-Scored ERM Generalization Error Bound of PFS). In imbalanced regression with bins partition $\mathbb{B}$, for any finite hypothesis space of predictions $\mathcal{H} = \{\hat{y}_1, \ldots \hat{y}_{|\mathcal{H}|}\}$, the transductive prediction error of the empirical risk minimizer $\hat{y}^{ERM}$, using the PFS estimator with estimated propensities $\tilde{P}(\tilde{P}_b > 0)$ and given training observations $O$ from $\mathcal{Y}$ with independent Bernoulli propensities $P$, is bounded by:*

$$R(\hat{y}^{ERM}) \leq \hat{R}_{\text{PFS}}(\hat{y}^{ERM}|\tilde{P}) + \underbrace{\frac{\Delta}{|\mathbb{B}|} \sum_{b=1}^{|\mathbb{B}|} |1 - \frac{P_b}{\tilde{P}_b}|}_{Bias} + \underbrace{\frac{\Delta}{|\mathbb{B}|}\sqrt{\frac{\log(2|\mathcal{H}|/\eta)}{2}}\sqrt{\sum_{b=1}^{|\mathbb{B}|} \frac{1}{\tilde{P}_b^2}}}_{Variance}. \quad (16)$$

*Proof.*

$$R(\hat{y}^{\text{ERM}}) \leq \hat{R}_{\text{PFS}}(\hat{y}^{\text{ERM}}|\tilde{P}) + \underbrace{R(\hat{y}^{\text{ERM}}) - \mathbb{E}_O[\hat{R}_{\text{PFS}}(\hat{y}^{\text{ERM}}|\tilde{P})]}_{Bias} + \underbrace{|\hat{R}_{\text{PFS}}(\hat{y}^{\text{ERM}}|\tilde{P}) - \mathbb{E}_O[\hat{R}_{\text{PFS}}(\hat{y}^{\text{ERM}}|\tilde{P})]|}_{Variance}$$

**Bias Term.** For the bias term:

$$R(\hat{y}^{\text{ERM}}) - \mathbb{E}_O[\hat{R}_{\text{PFS}}(\hat{y}^{\text{ERM}}|\tilde{P})]$$

$$= \frac{1}{|\mathbb{B}|} \sum_{b=1}^{|\mathbb{B}|} \frac{1}{|\mathbb{U}_b|} \sum_{i \in \mathbb{U}_b} \delta_i(y, \hat{y}^{\text{ERM}}) - \frac{1}{|\mathbb{B}|} \sum_{b=1}^{|\mathbb{B}|} \frac{1}{|\mathbb{U}_b|} \sum_{i \in \mathbb{U}_b} \frac{P_b}{\tilde{P}_b}\delta_i(y, \hat{y}^{\text{ERM}})$$

$$\leq \frac{1}{|\mathbb{B}|} \sum_{b=1}^{|\mathbb{B}|} \frac{1}{|\mathbb{U}_b|} \sum_{i \in \mathbb{U}_b} |1 - \frac{P_b}{\tilde{P}_b}|$$

$$= \frac{1}{|\mathbb{B}|} \sum_{b=1}^{|\mathbb{B}|} |1 - \frac{P_b}{\tilde{P}_b}|$$

**Variance Term.** For the variance term:

$$\mathbb{P}(|\hat{R}_{\text{PFS}}(\hat{y}^{\text{ERM}}|\tilde{P}) - \mathbb{E}_O[\hat{R}_{\text{PFS}}(\hat{y}^{\text{ERM}}|\tilde{P})]| \leq \epsilon) \geq 1 - \eta$$

$$\Leftarrow \mathbb{P}(\max_{\hat{y}_i} |\hat{R}_{\text{PFS}}(\hat{y}|\tilde{P}) - \mathbb{E}_O[\hat{R}_{\text{PFS}}(\hat{y}|\tilde{P})]| \leq \epsilon) \geq 1 - \eta$$

$$\Leftrightarrow \mathbb{P}(\bigvee_{\hat{y}_i} |\hat{R}_{\text{PFS}}(\hat{y}|\tilde{P}) - \mathbb{E}_O[\hat{R}_{\text{PFS}}(\hat{y}|\tilde{P})]| \geq \epsilon) < \eta$$

$$\Leftarrow \sum_{i=1}^{|\mathcal{H}|} \mathbb{P}(|\hat{R}_{\text{PFS}}(\hat{y}|\tilde{P}) - \mathbb{E}_O[\hat{R}_{\text{PFS}}(\hat{y}|\tilde{P})]| \geq \epsilon) < \eta$$

$$\Leftarrow 2|\mathcal{H}| \exp(-\frac{2\epsilon^2 |\mathbb{B}|^2}{\Delta^2 \sum_{b=1}^{|\mathbb{B}|} \frac{1}{|\mathbb{U}_b|} \sum_{i \in \mathbb{U}_b} \frac{1}{\tilde{P}_b^2}}) < \eta$$

$$\Leftrightarrow 2|\mathcal{H}| \exp(-\frac{2\epsilon^2 |\mathbb{B}|^2}{\Delta^2 \sum_{b=1}^{|\mathbb{B}|} \frac{1}{\tilde{P}_b^2}}) < \eta$$

, where $2|\mathcal{H}| \exp(-\frac{2\epsilon^2 |\mathbb{B}|^2}{\Delta^2 \sum_{b=1}^{|\mathbb{B}|} \frac{1}{|\mathbb{U}_b|} \sum_{i \in \mathbb{U}_b} \frac{1}{\tilde{P}_b^2}}) < \eta$ holds by (Lemma1). Solving the last line for $\epsilon$ concludes

that with probability $1 - \eta$, $|\hat{R}_{\text{PFS}}(\hat{y}^{\text{ERM}}|\tilde{P}) - \mathbb{E}_O[\hat{R}_{\text{PFS}}(\hat{y}^{\text{ERM}}|\tilde{P})]| \leq \frac{\Delta}{|\mathbb{B}|} \sqrt{\frac{\log(2|\mathcal{H}|/\eta)}{2}} \sqrt{\sum_{b=1}^{|\mathbb{B}|} \frac{1}{\tilde{P}_b^2}}$

By combining the bound for bias term and variance term, we reach the stated results. $\qquad\square$

# B EXPERIMENTAL DETAILS

## B.1 KERNEL FUNCTIONS

**Linear Kernel.** Linear Kernel is a commonly used kernel function, which can easily probe the linear relationship between input datum.

$$k_{\boldsymbol{\theta}}^{\text{LIN}}(\boldsymbol{x}_1, \boldsymbol{x}_2) = \boldsymbol{\theta}_1^2 + \boldsymbol{\theta}_2^2 \boldsymbol{x}_1^\top \boldsymbol{x}_2,$$

where $\boldsymbol{\theta}$ is the parameter learned during training.

**Radial Basis Kernel.** Radial Basis Kernel is also referred to as Gaussian Kernel or Squared Exponential Kernel, a stationary kernel derived from the squared Euclidean distance between two inputs.

$$k_{\boldsymbol{\theta}}^{\text{RBF}}(\boldsymbol{x}_1, \boldsymbol{x}_2) = \boldsymbol{\theta}_1^2 \exp(-\frac{\|\boldsymbol{x}_1 - \boldsymbol{x}_2\|^2}{2\boldsymbol{\theta}_2^2}),$$

where $\boldsymbol{\theta}$ is the parameter learned during training.

The computations of GP and PFS-PITC with kernel functions listed above are implemented on *PyTorch*(Paszke et al. (2019)).

## B.2 METRICS

In this subsection, we briefly clarify the metrics used to evaluate the performance of our experiments.

**Mean Squared Error(MSE).** MSE measures the average of the squares of the errors, which are the differences between predicted and actual values. It gives more weight to larger errors, making it particularly sensitive to outliers. A lower MSE indicates better model performance.

$$\text{MSE} = \frac{1}{n} \sum_{i=1}^{n} (y_i - \hat{y}_i)^2,$$

where $y$ and $\hat{y}$ denotes ground truth and model prediction values respectively.

**Mean Average Error(MAE).** MAE represents the average absolute differences between predicted and actual values. Unlike MSE, it treats all errors equally, providing a straightforward interpretation of model accuracy. A smaller MAE signifies better predictive performance.

$$\text{MAE} = \frac{1}{n}\sum_{i=1}^{n}|y_i - \hat{y}_i|.$$

**Negative Log Likelihood(NLL).** NLL quantifies how well the predicted probability distribution aligns with the observed data. It is commonly used in probabilistic models, where a lower NLL indicates a better fit of the model to the data.

$$\text{NLL} = \frac{1}{2}\log|\boldsymbol{K}_N + \boldsymbol{\Sigma}| + \frac{1}{2}\boldsymbol{y}^{\top}(\boldsymbol{K}_N + \boldsymbol{\Sigma})^{-1}\boldsymbol{y} + \frac{N}{2}\log(2\pi).$$

**Correlation Coefficient(CORR).** CORR measures the correlation coefficient between predicted and actual signals, making it a common metric for evaluating time series prediction. It validates the relationship between signals, where a higher CORR indicates better prediction accuracy of the time series signals.

$$\text{CORR} = \frac{1}{n}\sum_{i=1}^{n}\frac{\sum_{t}(y_{it} - \hat{\mathbb{E}}[y_i])(\hat{y}_{it} - \hat{\mathbb{E}}[\hat{y}_i])}{\sqrt{\sum_{t}(y_{it} - \hat{\mathbb{E}}[y_i])^2(\hat{y}_{it} - \hat{\mathbb{E}}[\hat{y}_i])^2}}.$$

### B.3 SYNTHETIC DATASET

The synthetic regression task considers the following ground-truth function with observation noise:$f(x) = \exp(x) * \cos(2\pi x) * (2 + \epsilon), \epsilon \sim \mathcal{N}(0,1)$. Training inputs are gathered from the combination of two separate distributions on input variables:$\mathcal{D} = \{(x_i, y_i)\} \cup \{(x'_j, y'_j)\}, x_i \sim \mathcal{N}(0,1), x'_j \sim \mathcal{N}(2,1)$. Test inputs are gathered from a uniform distribution and are free of observation noise:$\mathcal{D}_{\text{test}} = \{(x_i^*, y_i^*)\}, x_i^* \sim U(2.5, 3.5), \epsilon = 0$. This sampling process generates 200 training samples and 100 test samples with interleaved distributions to simulate training on an imbalanced dataset while testing with balanced demands.

We establish Gaussian Processes (GP), local GP, and PFS-PITC based on the RBF kernel, training them with the Adam optimizer at a learning rate of $3e - 3$ over 1,000 iterations. For PFS-PITC, we adopt a bin number of 80 to maximize its performance improvement. When implementing local GP, we set the number of clusters to 4, aiming to enable its adaptation to varying noise amplitudes. The outcomes of this experiment are presented in Figure 3, which illustrates the training and test samples along with the mean ± standard error curves of the three algorithms for comparison.

### B.4 REGRESSION DATASET

In the regression dataset experiment, we first preprocess the input features and split the dataset to simulate sampling imbalance. We eliminate duplicate observations of input variables and apply min-max normalization to mitigate potential distribution shifts. Next, we create 100 equal-distance intervals for the sample targets and corresponding variables to assess target distribution. Instances within each bin are allocated to the training, validation, and test datasets in sequence until the quota is met. The quota is set at the 90th percentile of the sample size for each bin, ensuring label balance in the test set. After this process, the sizes of the training, validation, and test sets are 4,656, 2,289, and 2,582 for the *CCPP* dataset, and 4,656, 2,289, and 2,582 for the *CCS* dataset.

To evaluate the influence of feature extractor on different algorithms, we employ the feature extractor introduced in Finn et al. (2017) on *CCS* dataset, which consists of a two-layer MLP with ReLU activation that doubles the dimensionality of the input features. The Adam optimizer is again utilized to train the model over 500 iterations, with a batch size of 512 samples. A learning rate of $1e - 3$ is applied to the feature extractor, while a learning rate of $3e - 3$ is used for the GP parameters.

Most algorithms in this experiment synthesize artificial samples to alleviate sampling imbalance, implemented on the training set prior to feature extraction. An exception is FDS, which applies

smoothing to latent features for performance enhancement. Utilizing the source code of FDS (Yang et al. (2021)), we implement it to operate during the training iteration of GP. For PFS-PITC, the smoothing operation is performed using a Gaussian kernel with a size of 5 and an update rate of 0.05. We test three numbers of bins on each dataset to validate the stability of PFS-PITC concerning the partitioning of bins.

The experiment for each algorithm is repeated five times to provide statistical information for stability assessment. We record performance metrics, including MAE, MSE, and NLL, to demonstrate the impact of each algorithm on both regression output and parameter optimization of the Gaussian process model.

### B.5 TIME SERIES PREDICTION

In the preprocessing stage, we apply maximum normalization to ensure numerical stability. The dataset is then split into training (60%), validation (20%), and test (20%) sets in chronological order. The feature extractor combines CNN and MLP architectures. An initial 2D convolutional layer with a kernel size of 6 captures local temporal patterns from the input data, followed by a ReLU activation layer and a dropout layer (dropout rate = 0.2). The two-layer MLP is configured with intermediate and output layer dimensions set to 512 and 256, respectively. For the Gaussian Process, we utilize a mixture of linear and RBF kernels. A regularization coefficient of $1e-4$ is added to the diagonal of the kernel matrix to prevent floating-point calculation errors that may lead to a non-positive definite kernel matrix. During the training process, we employ the Adam optimizer to train each model for 500 epochs, using a learning rate of $3e-3$ for GP parameters and $1e-3$ for the feature extractor. In the calculation of PFS-PITC, the smoothing operation is applied using a Gaussian kernel with a size of 3 and an update rate of 0.05. We present the test results for PFS-PITC with three different numbers of bins to examine the impact of bin partitioning on time series prediction. Additionally, three look-ahead horizons $\{3, 6, 12\}$ are tested on each dataset to assess the prediction stability of each algorithm.