# OpenReview forum: "Probabilistic Feature Smoothed Gaussian Process For Imbalanced Regression"
_ICLR.cc/2025/Conference — ICLR 2025 Conference Withdrawn Submission_

### Official Review · Reviewer_5sQe · 2024-10-20

**Soundness:** 2
**Presentation:** 1
**Contribution:** 2
**Rating:** 3
**Confidence:** 4

**Summary:**

In fitting Gaussian process regression models, there is a problem where the predictive accuracy of the fitted model decreases if the values taken by the response variable are biased. To address this issue, this paper proposes to divide the space of the response variable equally and calculate the empirical mean and empirical covariance matrix of the features extracted from the explanatory variables corresponding to each division. By weighting these means and covariance matrices using the similarity of the response variable (expressed with a kernel function) and averaging them, new normal distributions corresponding to each division are constructed. Then features are sampled from these distributions to serve as inducing points.

**Strengths:**

The experimental results compared with other methods developed to address imbalance seem to be favorable.

**Weaknesses:**

The novelty and motivation of the paper are unclear.
First, regarding the novelty.
The fundamental idea of this paper is similar to Sliced Inverse Regression (SIR), where the data is divided based on the space of the response variable, and the statistics of the explanatory variables in each division are utilized. Therefore, SIR should be cited as a key related work:
Li, Ker-Chau. 1991. “Sliced Inverse Regression for Dimension Reduction.” Journal of the American Statistical Association 86 (June): 316–27.

Specifically, there are similar method, which applies kernel weights to each data point based on the value of the response variable to perform smoothing:
Zhu, Y., & Fang, K. T. (1996). Asymptotics for kernel estimate of sliced inverse regression. Annals of Statistics, 24(3), 1053-1068.

Local SIR, which emphasizes data points with similar response values to estimate the dimension reduction directions:
Chen, C. H., & Li, K. C. (1998). Can SIR be as popular as multiple linear regression?  Statistica Sinica, 8(2), 289-316.

and Weighted SIR, which incorporates information from adjacent slices by weighting the mean vector of each slice.:
Xia, Y., & Härdle, W. (2006). Semi-parametric estimation of partially linear single-index models. Journal of Multivariate Analysis, 97(5), 1162-1184.

I understand the above mentioned papers are for supervised dimension reduction and the submitted paper is for Gaussian process fitting, but the used technique is quite similar. The relationship between these methods and the positioning of the proposed method should be clearly explained.

I will comment on the motivation in "Question" later.

The presentation of the paper should be considerably improved.
There are so many presentation issues that they cannot all be pointed out. I'll point out few of them:
- In regression problems, it is not as common to refer to the response variable y as a "label" compared to classification problems. In classification tasks, the term "label" is frequently used to denote discrete classes (categories). However, in regression tasks, where the response variable y takes continuous values, terms like "target" or "response variable" are more commonly used. That being said, using the term "label" in regression problems is not entirely incorrect. However, for greater precision, using terms like "target" or "response" is generally more appropriate in regression tasks.
- Please carefully read the example (template) of the ICLR paper. "The table number and title always appear before the table."
- On page 1, abbreviations such as SoR, DTC, FITC, and PITC are used without explanation, despite their first appearance. Similarly, on page 2, abbreviations like DIR, LDS, and FDS are used right from the start without being spelled out. While these are spelled out later, they should be spelled out upon their first appearance. Relatedly, this paper contains too many undefined notations and terms, or notations and terms that are used before being defined. Notations such as \( K_{N,M}, K_M, \bar{f}, K_N, Q_N, f_{B_s} \) in Eqs. (1), (2), and (3) should not be used without definition. Given that this is a Gaussian process paper, one could generously assume that \( k \) represents the kernel function, but it is still inappropriate from a writing perspective to use symbols like \( k \) and \( Q \) without defining them.
- While I understand that \( \mathcal{Y}_n \) represents the partitioning of the response variable into intervals, what does \( |\mathcal{Y}_b| \) in the lower part of p.3 represent?
- 'blockdiag' is not defined.
- The summation in Eq. (8) is used inaccurately.
- Why is the dimension of \( \mathbf{f} \) in Eq. (11) \( B \)?
- Right below Eq. (11), it says 'log marginal likelihood (NLL)', but why is the abbreviation for log marginal likelihood 'NLL'?
- In Algorithm 1, \( \mathbb{B} \) is referred to as the Bin index, but considering the definition of \( \mathbb{B} \), calling it an 'index' seems inappropriate.

**Questions:**

1. Please provide a detailed discussion of the proposed method's relation to existing SIR-like approaches in the statistics literature and explicitly state its difference and advantage over existing ones.

2. Please clarify the motivation of the proposed approach on two aspects:
i) what happens when the response variable has an imbalanced distribution?
ii) why the neural network for feature extraction is needed.
While it is indeed plausible that an imbalanced response variable could lead to some issues, the specific mechanisms and the exact nature of the problems that arise are not clearly explained. Furthermore, the use of neural networks for feature extraction does not appear to be essential to the core of this research.

---

### Official Review · Reviewer_MvVS · 2024-11-02

**Soundness:** 2
**Presentation:** 2
**Contribution:** 1
**Rating:** 3
**Confidence:** 4

**Summary:**

Gaussian Processes (GPs) often struggle with imbalanced regression, where uneven target label distributions negatively impact performance. This paper addresses this issue by proposing a new method, Probabilistic Feature Smoothed Partially Independent Training Conditional Approximation (PFS-PITC), aimed at enhancing GP accuracy in the presence of imbalanced data. By extracting features across equal intervals of target labels and applying kernel smoothing, PFS-PITC mitigates gaps in sampling density and leverages information from nearby labels. The authors evaluate PFS-PITC on various imbalanced datasets, demonstrating its effectiveness in improving GPs' robustness and applicability for complex, real-world data tasks.

**Strengths:**

The paper is well-structured and well-written. The problem is clearly defined, and the authors effectively address recent, relevant related works. Label imbalance is a significant issue in prediction problems, and the improvements proposed in this framework may present a valuable contribution.

**Weaknesses:**

* Contribution: The paper’s contribution is limited, as it builds on the PITC framework, which is not novel. Simply adding a calibration layer to PITC may lack impact, particularly as this layer could potentially be integrated with more current methods.

* Feature Distribution Smoothing (FDS): This layer is insufficiently explained in the paper. Further discussion is needed to clarify how FDS substantially improves the results and what mechanisms make it effective.

* Computational Cost: Inducing point-based methods remain computationally expensive, a limitation that the paper does not adequately address. Including a discussion on computational complexity would strengthen the paper’s rigor.

* Quality of Estimation: The accuracy of estimation with inducing points is highly sensitive to the quantity and selection process of these points. However, the paper does not sufficiently consider these limitations.

* Hyperparameter Tuning for Bins: The paper would benefit from a discussion on selecting the number of bins and initializing this hyperparameter. It remains unclear whether increasing the number of bins would enhance result quality.

**Questions:**

The label space is divided into equidistant intervals. What happens if one interval has no labels in an imbalanced dataset? Is the model still stable when there are empty bins?

---

### Official Review · Reviewer_hgSD · 2024-11-03

**Soundness:** 1
**Presentation:** 2
**Contribution:** 1
**Rating:** 1
**Confidence:** 4

**Summary:**

In this paper, a method called PFS-PITC is proposed as a Gaussian Process model for imbalanced regression problems. The proposed method is based on two existing methods: PFS and PITC. PFS is designed for imbalanced regression and uses an approach based on kernel smoothing of the response variable, while PITC divides the dataset into groups and learns a GP for each group. For grouping in PITC, this study adopts an approach that divides the response variable into equal intervals. The authors claim, based on theoretical analysis and numerical experiments, that the proposed method outperforms existing methods.

**Strengths:**

1. I fully agree with the authors' point that modeling an imbalanced regression dataset using a single Gaussian Process Model poses issues in terms of computational cost, approximation capability, and stability.

2. The problem of properly handling imbalanced data is a frequent and important issue in real data analysis, yet effective methodologies are lacking, making the problem addressed in this paper worthwhile studying.

**Weaknesses:**

1. First, there seems to be a need for a clear discussion on the necessity of modeling imbalanced regression data, as addressed in this paper, using a Gaussian Process model. If I encountered such a data analysis problem, I would likely choose another regression model. For instance, regression models based on tree structures, such as random forest or boosting, are known to behave relatively robustly even when the response variable takes extreme values.

2. I am skeptical about whether it is appropriate to use only the values of the response variable for data partitioning in PITC. With this type of partitioning, regions with similar response variable values but significantly different input values would be represented by a single Gaussian Process model, which seems inappropriate.

3. In the numerical experiments, the performance of the proposed method is overwhelmingly better than that of other baseline methods; however, the experimental setup is overly complex and specific, making it difficult to properly interpret the results. I have the following concerns regarding the experimental settings among others:

   a) Why were the combined cycle power plant and concrete compressive strength datasets chosen? I am curious whether it was confirmed that these datasets represent imbalanced regression problems.

   b) How were the hyperparameters selected? Setting the GP hyperparameters the same for both the single GP model and PITC is clearly inappropriate—if optimal hyperparameters were specifically selected for the proposed method, that would be unfair.

   c) Why were the response variables divided into 100 intervals? Was it reasonable to make that many partitions? Modeling with 100 local GPs seems likely to result in highly overfitted results.

4. The proposed method appears to be merely a simple combination of existing methods, PITC and PFS, lacking sufficient originality to warrant acceptance at a top-tier conference like ICLR.

**Questions:**

It was unclear whether the theoretical analysis statements were specific to the proposed PFS-PITC or applicable to PFS in general. I would like an explanation of how local modeling in PITC affects the statements of the lemmas and theorems.

---

### Official Review · Reviewer_qiV9 · 2024-11-03

**Soundness:** 3
**Presentation:** 3
**Contribution:** 2
**Rating:** 5
**Confidence:** 4

**Summary:**

This paper studies the problem of imbalanced regression where the real line has a non-uniform marginal distribution of target values in the setting of using Gaussian Processes for function approximation. It proposed to address this issue by assuming that the learned function is smooth so that input features from training examples in more densely populated target value regions can be "borrowed" to interpolate more appropriate inputs. Results on real-world datasets are promising.

**Strengths:**

The paper gives an excellent introduction into approximate Gaussian processes using inducing points and also presents the real-world datasets in detail; I really enjoyed reading this. The paper is well written and easy to follow.

**Weaknesses:**

It is not clear (and made clear) what (implicit) assumptions are made for the smoothing to work. Also, I did not understand how Section 4.3 relates to the paper; these are worst-case bounds for an expected loss function approach on p(y|x) yet using Gaussian processes means that we already make a distributional assumption on p(y|x). It would have been better to study the relationship between p(x) and p(y|x) that has to be fulfilled for the approach to work well.

**Questions:**

* In line 108, $Q$ is indexed by both $N$ and $M$ but in (1) - (3) it's only indexed by $N$ (I assume that one index means that $A$ = $B$)
* In line 161, how can $|Y_b|= C$ if $Y_b$ is an interval on the real line? Isn't that a set with an infinite number of elements (by definition)? I think what you mean is $y^{(b)} - y^{(b-1)}=C$
* On line 212, $\mu$ should be $\mu_b$
* Line 338: Shouldn't this really be a mixture of two Gaussian for $p(x)$?
* Figure 3: I find it very difficult to make out why the green line is a better fit; is the blue line the true function (without noise)? If so, why does the test set not have an observation noise?

---

### Note · Authors · 2024-11-22

**Comment:**

We would like to express our gratitude to the reviewers for their time and constructive feedback. After careful consideration, we have decided to withdraw this submission and will focus on improving the paper based on the valuable insights we have received.

**Withdrawal Confirmation:**

I have read and agree with the venue's withdrawal policy on behalf of myself and my co-authors.